# The Photothermal Conversion and UV Resistance of Silk Fabrics Being Achieved through Surface Modification with C@SiO_2_ Nanoparticles

**DOI:** 10.3390/molecules28247970

**Published:** 2023-12-06

**Authors:** Gang Deng, Lu Yao, Mingzhao Chen, Yuanyuan Yang, Song Lu, Guohua Wu

**Affiliations:** 1College of Biotechnology and Sericultural Research Institute, Jiangsu University of Science and Technology, Zhenjiang 212100, China; jayderek@foxmail.com (G.D.);; 2Huangshan Kehong BioFlavor Co., Ltd., Huangshan 245200, China

**Keywords:** C@SiO_2_ nanomaterials, photothermal conversion, UV resistance, clean production

## Abstract

With the improvement in people’s living standards, the development and application of smart textiles are receiving increasing attention. In this study, a carbon nanosurface was successfully coated with a SiO_2_ layer to form C@SiO_2_ nanomaterials, which improved the dispersion of carbon nanomaterials in an aqueous solution and enhanced the absorption of light by the carbon nanoparticles. C@SiO_2_ nanoparticles were coupled on the surface of silk fabric with the silane coupling agent KH570 to form C@SiO_2_ nanosilk fabric. The silk fabric that was subjected to such surface modification was endowed with a special photothermal function. The results obtained with scanning electron microscopy (SEM), energy dispersive spectrometer (EDS), and infrared spectroscopy (FTIR) showed that C@SiO_2_ nanoparticles were successfully modified on the surface of the silk fabric. In addition, under the irradiation of near-infrared light with a power of 20 W and a wavelength of 808 nm, the C@SiO_2_ nanosilk fabric experienced rapid warming from 23 °C to 60 °C within 30 s. After subjecting the functional fabric to hundreds of photothermal experiments and multiple washes, the photothermal efficiency remained largely unchanged and proved to be durable and stable. In addition, the thermogravimetric (TG) analysis results showed that the C@SiO_2_ nanoparticles contributed to the thermal stability of the silk fabric. The UV transmittance results indicated that C@SiO_2_ nanofabric is UV-resistant. The silk modification method developed in this study is low-cost, efficient, and environmentally friendly. It has some prospects for future applications in the textile industry.

## 1. Introduction

Silk, obtained from silkworms, is a natural polymer material. Silk fabrics are characterized by a unique pearl-like luster, breathability, high moisture absorption, and smooth hand feel. In addition, silk fabrics are also strong and ductile, highly biocompatible, biodegradable, and environmentally stable [1]. Although silk fabrics excel in many aspects, they also have some drawbacks: silk fabrics are prone to yellowing and wrinkling, which seriously affect their esthetics and longevity. Additionally, the poor UV resistance and heat resistance of silk fabrics pose some problems [2]. To overcome these drawbacks, the current performance of silk fabrics should be enhanced through modification to achieve special properties. As such, researchers have focused on the modification of silk fabrics and the functional finishing of fabrics. Through continuous technological innovation, research, and development, we hope to further expand the applications of silk fabrics [3,4].

With global energy shortages and environmental pollution becoming increasingly serious, people are increasingly concerned about low carbon energy savings and environmental protection. With respect to indoor life, thermal comfort is achieved with electrical devices such as air conditioners and heaters; yet, these devices consume large amounts of energy. Therefore, a smart wearable fabric should be developed that can protect the body from the cold in cold environments but can also absorb the energy from the sunlight and turn it into heat or use reflected far-infrared radiation from the human body for heating and insulation [5,6], thus providing protection against the cold in extremely harsh climatic conditions. To achieve thermal comfort for the human body while reducing energy consumption for warmth, personal thermal management is an effective solution [7,8,9]. Traditional smart textiles include the photovoltaic effect [10], electrothermal conversion [11], and photothermal conversion [12]. Solar energy, as an abundant, safe, and sustainable energy source, is widely used in photothermal conversion technology and can provide a convenient and reliable solution for personal temperature regulation.

Solar energy is an inexhaustible and renewable resource, which is why photothermal textiles have been extensively studied. Photothermal conversion materials absorb near-infrared light and generate heat through plasma resonance thermal effects, semiconductor nonradiative chirality, and molecular thermal vibrations, resulting in rapid localized warming [13,14]. The general photothermal conversion nanomaterials are classified as metallic [15], semiconductor [16], and carbon-based nanomaterials [17]. Among them, carbon-based nanomaterials are simple to prepare, low-cost, and have a good heat transfer capacity and excellent chemical and physical properties [18]. Therefore, they can be used as photothermal conversion materials [19], anti-UV materials [20], and electromagnetic shielding materials [17,21]. Because carbon-based nanomaterials have SP^2^ and SP^3^ hybrid orbitals with very close energy levels and π-electron clouds, they show excellent photothermal conversion performance and high efficiency due to their strong absorption of near-infrared light with a wide range of visible light absorption wavelengths [22,23]. Therefore, they can be used as a photothermal conversion material for wearable smart fabrics. In addition, silica is considered a promising material because of its low cost, chemical inertness, good mechanical properties, thermal stability, and its nonpolluting nature [24]. Nanosilica has very low photocatalytic activity and high UV and IR shielding, as well as reflecting abilities [25]. Therefore, it is suitable for a wide range of promising applications in textile surface modification.

To combine the advantages of carbon and silica nanoparticles, in this study, a carbon–silicon composite photothermal nanomaterial was prepared by coating a layer of silica on carbon nanoparticles. As shown in Figure 1a, carbon nanoparticles have a high surface energy and are prone to agglomeration and precipitation in solution. After modification with PVP, the carbon nanoparticles were less agglomerated and better dispersed in a solution, and the affinity of the carbon nanoparticles for the SiO_2_ precursor was strengthened [26]. In this study, a modified Stober method was used to form silica layers on the surface of carbon nanoparticles under alkaline conditions to form core–shell nanostructures [27,28]. The employed coating process substantially increased the dispersion of carbon nanoparticles in solvents such as water and ethanol [29]. The C@SiO_2_ nanoparticles bonded to the silk surface using a silane coupling agent (KH570) to form functional fabrics. The C@SiO_2_ silk fabrics exhibited excellent photo-thermal conversion and UV resistance properties. In addition, a layer of silica was coated on the surface of the carbon nanomaterials to improve the carbon nanodispersion and absorption of light, enhancing the safety of the wearable fabric. The photothermal and UV resistance of silk fabrics enables the use of silk for smart wearable textiles.

## 2. Results and Discussion

When the carbon nanoparticles and C@SiO_2_ nanoparticles were observed through a transmission electron microscope (TEM), we observed that the surface of the carbon nanomaterials, as shown in Figure 1a,b, was smooth, showing a regular spherical shape. When a layer of SiO_2_ was modified on the surface of the carbon nanomaterials, we observed a transparent layer of SiO_2_ wrapped around the surface of the carbon nanomaterials, as shown in Figure 1c,d. Furthermore, the EDS mapping results depicted in Figure 1e demonstrate that the surface of C@SiO_2_ nanoparticles possesses uniformly distributed Si and O elements in addition to C. This confirms that the silica is effectively encapsulated on the carbon nanosurface. Figure 1f showcases the XRD patterns of the carbon nanoparticles, which feature characteristic peaks at 26.5° and 42.8°, corresponding to the (002) and (100) characteristic peaks of graphitized carbon. It is evident that the morphology of the carbon nanoparticles remained unaltered upon encapsulation with a silica layer. However, the intensity of the (002) peak exhibited a slight increase, indicating the successful encapsulation of silica on the surface of the graphitized carbon nanoparticles.

Carbon and C@SiO_2_ nanoparticles were dispersed in aqueous solution and the UV−Vis−NIR absorption spectra were measured, as shown in Figure 2a. The results showed that the absorbance of the C@SiO_2_ nanoparticle dispersions was stronger than that of the carbon nanoparticle dispersions at the same mass concentration due to the rich hydrophilic hydroxyl functional groups on the surface of the C@SiO_2_ nanoparticles which allowed the C@SiO_2_ nanoparticles to be more easily dispersed in aqueous solution. In addition, C@SiO_2_ nanoparticles can reduce light reflection and increase light absorption [30,31]. To verify the stability of the C@SiO_2_ nanoparticles in aqueous solution, we measured the absorbance of the carbon nanoparticles and C@SiO_2_ nanoparticles dispersed in aqueous solution at a wavelength of 808 nm at different times, as shown in Figure 2b. We found that the carbon nanoparticles completely precipitated after only 3 h in aqueous solution, as shown in Figure 2c, and the absorbance of the supernatant was close to zero. In contrast, the luminosity of the C@SiO_2_ nanoparticles remained almost unchanged over 7 days. This indicates that the C@SiO_2_ nanoparticles were more dispersed in aqueous solutions and did not precipitate or coalesce. The zeta potentials of carbon nanoparticles and C@SiO_2_ nanoparticles dispersed in aqueous solution were also measured separately, as in Figure 2d,e. The results show that the negative charge carried by the surface of carbon nanoparticles experienced enhancement following silica wrapping. This means that, in aqueous solution, the carbon nanoparticles are more stable and less prone to agglomeration and precipitation. This may be due to the presence of the silica layer, which hinders the mutual attraction and accumulation of carbon nanoparticles. As such, these nanoparticles are more feasible for practical application in textile processes.

From the scanning electron microscopy (SEM) results shown in Figure 3a we found that the surface of the untreated silk fabric was smooth and flawless; after the C@SiO_2_ nanotreatment, the surface of the C@SiO_2_ nanosilk fabric was rough, a phenomenon which indicated that the C@SiO_2_ nanoparticles were successfully modified on the silk surface, possibly as a result of the condensation reaction of the silane coupling agent KH570 in grafting the C@SiO_2_ nanoparticles on the silk fabric surface [32]. By comparing the energy dispersive spectrometer (EDS) results of the untreated silk fabric with those of the C@SiO_2_-nanoparticles-treated silk fabric, as shown in Figure 3b, we found that the content of elemental silicon in the untreated silk fabric was 0%, whereas that in the C@SiO_2_-nanoparticles-treated silk fabric was 29.59%, a fact which proved that the C@SiO_2_ nanoparticles were successfully modified on the silk surface. In addition, Figure 3b shows that elemental Si was uniformly distributed on the silk surface, indicating that the C@SiO_2_ nanoparticles were uniformly distributed on the silk surface.

As shown in Figure 4a, the carbon nanoparticles mainly showed the antisymmetric -OH stretching vibration peak at 3420 cm^−1^. When the carbon nanoparticles were wrapped with a layer of SiO_2_ on the surface, we found that the antisymmetric -OH stretching vibration peak at 3420 cm^−1^ significantly increased compared to the IR spectra, and the new characteristic peaks at 1097 cm^−1^ and 468 cm^−1^ corresponded to the stretching and bending vibration of the silicon–oxygen bond, respectively. The strong absorption peak at 1097 cm^−1^ corresponded to the asymmetric vibration of the Si–O–Si in the sample. The absorption peak at 468 cm^−1^ corresponded to the bending vibration of Si–O [33]. The IR results demonstrate the successful wrapping of a layer of SiO_2_ on the carbon nanosurface. The results in Figure 4b show that the characteristic peaks of the random coils in amide I and the β-folded structure in the amide II band, both of which belong to the main characteristic peaks of silk proteins, appeared at 1648 cm^−1^ and 1527 cm^−1^, respectively [34]. The silk fabric modified with the silane coupling agent KH570 showed an asymmetric -CH_3_ stretching vibration peak at 2927 cm^−1^, C–O–C absorption peak at 1300 cm^−1^, and C=O stretching vibration peak at 1722 cm^−1^ [35,36]. The silk-modified nanoparticles then showed a C–O stretching vibration peak linked to an alkyl group at 1037 cm^−1^. The Si–O–Si bonds may have formed due to the hydrogen bonding between the Si–OH generated by the hydrolysis of Si–CH_3_ in KH570 and the hydroxyl groups on the silica surface [37]. We found that C@SiO_2_ was modified on the silk surface with the silane coupling agent KH570 in the form of grafting through chemical bonding.

The untreated silk fabric, the silk fabric treated with the silane coupling agent KH570, and the silk fabric cotreated with different concentrations of C@SiO_2_ nanoparticle solutions were each subjected to NIR light irradiation at a power of 20 W and a wavelength of 808 nm. The change in temperature of the silk fabric over time was observed via infrared thermography, and the results are shown in Figure 5a. Notably, the C@SiO_2_ silk fabric experienced rapid warming from 23 °C to 60 °C within 30 s, demonstrating its remarkably rapid heating ability under 808 nm NIR light. In addition, to determine the photothermal conversion of the C@SiO_2_ silk fabric under natural light, the samples were placed together under natural light. The temperature change in the silk fabric over time was observed with infrared thermography, and the results are shown in Figure 5b. The results showed that the temperature of the C@SiO_2_ nanosilk fabric increased from an outdoor temperature of 22 °C to 45 °C within 8 min, proving capable of a high photothermal conversion efficiency under sunlight. Based on the results of the photothermal data obtained in near-infrared and natural light, we found that the difference in the photothermal effect of the silk fabrics with different nanoparticle concentrations was minimal. This occurred because the C@SiO_2_ nanomaterial is a highly efficient photothermal material, requiring only a small amount to produce a powerful photothermal effect. Therefore, the C@SiO_2_-nanomaterial-modified silk fabric has considerable potential as a functional fabric for photothermal conversion. Furthermore, by observing the photothermal efficiency of the C@SiO_2_ and carbon nanosilk fabrics under 808 nm NIR light, we found that the photothermal effect of both materials was strong, but that of the C@SiO_2_ silk fabric was stronger. This occurred because the C@SiO_2_ nanoparticles were more uniformly dispersed on the silk surface and did not agglomerate, resulting in a stronger photothermal effect. In other words, the C@SiO_2_-nanomaterial-modified silk fabric provides additional advantages and is more suitable as a material for functional fabrics for photothermal conversion.

Next, to test the reproducible photothermal performance of the C@SiO_2_ silk fabric, we placed the latter under 808 nm NIR light, repeatedly warmed and cooled it, and observed its temperature change over time, with the results shown in Figure 5d. The experimental results showed that the C@SiO_2_ silk fabric quickly warmed and reached equilibrium after 30 s. The results of 100 repeated photothermal conversion experiments showed that the material warmed up from 23 °C to approximately 60 °C in 30 s and demonstrated good photothermal stability: its photothermal efficiency was not reduced by repeated use.

To test the effect of the number of washes on the experimental effect of the photothermal conversion of the C@SiO_2_-nanoparticle-treated silk fabric, different numbers of washes were applied, and a comparative experiment was conducted. with the results shown in Figure 6a,b. The experimental results showed that the photothermal conversion effect of the C@SiO_2_-treated silk fabrics differed little after 0, 10, 30, and 50 washes under either 808 nm NIR or sunlight, which proved that the bond between the C@SiO_2_ nanoparticles and silk fabrics was strong and not easily broken.

The thermal stability of the fabrics was effectively assessed with thermogravimetric (TG) and derivative thermogravimetric (DTG) analyses, as shown in Figure 7a,b. In general, silk has a small mass loss in the range of 40–200 °C, mainly due to the evaporation of water; in the range of 320–480 °C, silk fabrics show a large mass loss, which is due to their thermal degradation [3]. Therefore, we concluded from the readings that the thermal degradation temperature of the untreated silk fabric was 324 °C and that of the C@SiO_2_-nanoparticle-treated silk fabric was 320 °C. We noted a slight acceleration in the decomposition, possibly occurring due to the ability of C@SiO_2_ nanoparticles to increase the absorption of heat by the fabric. Furthermore, by comparing the residual mass, we found that the percentage of residual mass of the C@SiO_2_ nanomodified silk fabric was 33.8%, which was 7.4% higher than that of the untreated silk fabric. This also indicates that the silk surface was successfully modified with C@SiO_2_ nanoparticles. Overall, these results indicate that C@SiO_2_ nanoparticles can effectively improve the thermal stability of silk.

To verify the UV resistance of the C@SiO_2_ silk fabric, the UV transmittance of the fabric was measured, as shown in Figure 8, and the UV protection factor (UPF), UVA, and UVB transmittance of the fabric were calculated. The results are shown in Table 1. The UPF of the untreated silk fabrics was only 38.57. Different concentrations of C@SiO_2_-nanoparticle-treated silk fabrics had UPF values of up to 165.76, indicating that the C@SiO_2_ silk fabrics had a certain UV resistance. This performance mainly depended on the UV absorption capacity of the carbon nanomaterials and the UV reflection capacity of SiO_2_. Therefore, C@SiO_2_ silk fabric is a promising functional fabric.

As shown in Table 2, C@SiO_2_ silk fabric is compared with other heating fabrics in terms of heating performance. It can be seen that C@SiO_2_ silk fabric has good performance of rapid heat generation and has significant advantages over other heating fabrics. Therefore, C@SiO_2_ silk fabric has a good prospect in future smart fabric applications.

## 3. Experimental

### 3.1. Materials

Silk fabric (19 mm, density: 114 g/m^2^) was purchased from Ningbo Yunling Textile Trading Co. (Ningbo, China). Carbon nanomaterials (particle size: 50 nm, purity: 99.99%) were purchased from Suzhou Beasley New Materials Co. (Suzhou, China). We purchased 3-(trimethoxysilyl)propyl methacrylate (KH570), polyvinylpyrrolidone (PVP), and tetraethyl silicate (98%) (TEOS) from Aladdin Biochemical Technology Co. (Shanghai, China); ammonia (25%) was purchased from Xi long Science Co. (Shantou, China). A plug-in thermal imager (312-Z5mini) was obtained from Shenzhen Chuang zhi Fei Technology Co., Ltd. (Shenzhen, China), and an 808 nm NIR LED lamp (20 W) was obtained from Fang pu Optoelectronics Co. (Shenzhen, China).

### 3.2. Preparation of C@SiO_2_ Nanoparticles

Firstly, 6 g of carbon nanoparticles was added to 600 mL of deionized water and sonicated and dispersed for 30 min; secondly, an equal amount of polyvinylpyrrolidone (PVP) was added and stirred for 12 h at 1500 rpm to allow sufficient dispersion. Thirdly, the product was centrifugally dried and again sonicated and dispersed into 1000 mL of anhydrous ethanol. A total of 22.4 mL of deionized water and 32 mL of ammonia solution at a concentration of 0.1 M was added and then well stirred. Fourthly, 92.8 mL of tetraethyl silicate (TEOS) was added and stirred in a water bath at 38 °C for 5 h. Finally, C@SiO_2_ nanoparticles were obtained by centrifugal drying, as shown in Figure 1a.

### 3.3. Preparation of C@SiO_2_ Silk Fabric

The silk fabric was boiled in 7 mg/mL Na_2_CO_3_ solution for 30 min, followed by repeated washing with deionized water. The dry, degummed silk fabric was cut into 4 × 4 cm^2^ pieces, and 10 mL of C@SiO_2_ nanodispersion solution with different concentrations (1.25 mg/mL, 2.5 mg/mL, 5 mg/mL, 10 mg/mL, or 20 mg/mL) was produced. The silk fabric was immersed in the solution and the nanoparticles were dispersed by ultrasonication. Then, 200 µL of silane coupling agent KH570 was added and shaken in a water bath at 80 °C for 30 min. The silk fabric was then removed and transferred to an oven and dried at 120 °C. By repeating this operation several times, a sufficient amount of C@SiO_2_ nanoparticles were successfully modified onto the surface of the silk. To remove the unbound nanoparticles, the dried silk fabric needed to be repeatedly rinsed with deionized water and put into the oven again to dry, as shown in Figure 1b.

### 3.4. Characterization

Transmission electron microscopy (TEM) was used to observe the nanomorphology. A subset of nanoparticles was dispersed in a volume of anhydrous ethanol solution and sonicated for 30 min, and then a microdrop of the solution was taken onto a copper grid using a pipette gun and dried in air. After drying, they were placed in transmission electron microscopy (TEM) for characterization. A physical phase analysis of the nanomaterials was conducted through X-ray diffraction (XRD). First, the nanoparticle samples were placed on slides and flattened with a glass slide using alcohol. We used a Cu target with Kα rays for the analysis. Technical abbreviations are explained upon first usage. The structure is logical with causal connections between statements and a clear organization of information. The scanning angle ranged from 10° to 80° with a scanning rate of 5°/min and a step angle of 0.02°. The absorption of the nanoparticles at different optical wavelengths was tested using UV–Vis–NIR absorption spectroscopy. Fixed amounts of carbon nanoparticles and C@SiO_2_ nanoparticles were dispersed in aqueous solution, separately, and sonicated for 30 min. The nanoparticle dispersion was then pipetted into a quartz cuvette with a pipette and their UV–Vis–NIR absorption spectra were measured. The carbon nanoparticles and C@SiO_2_ nanoparticles dispersions were added to separate beakers and sonicated for 30 min; the supernatants were then aspirated at different times to determine their UV absorbance values. The zeta potential of nanoparticles in aqueous solution was determined using a particle size potentiostat (Zetasizer Nano ZS90, Malvern, UK). An equal amount of sample was dissolved into an equal amount of deionized water, and the supernatant was collected after sonication for 20 min for the test. The zeta potential was measured in scanning mode with 21 scans, the frequency range was 0–500 Hz, and the voltage range was ±200 V. The zeta potential of the nanoparticles in aqueous solution was determined using a particle size potentiostat in aqueous solution. The chemical structure of the nanoparticles and the silk fabric were observed using infrared spectroscopy (FTIR). A trace sample was mixed with a small amount of potassium bromide and then dried in an oven, ground, and pressed to form very thin slices for infrared spectroscopy measurements in a dry environment. The infrared spectra were measured in the 4000 cm^−1^ to 400 cm^−1^ band with a resolution of 4 cm^−1^ and 256 cumulative scans. Scanning electron microscopy (SEM) was used to analyze the surface morphology of the silk fabrics. In addition, the surface of the silk fabric was scanned for elements using energy dispersive spectrometer (EDS) (Hitachi SU-70, Tokyo, Japan). The thermal stability of the fabrics was tested using thermogravimetry (TG). The silk fabric was cut up and tested for thermal stability using a thermogravimetric analyzer STA-2500 NE-TZSCH, Mettler Toledo, Giessen, Germany, under a nitrogen atmosphere. The sample mass was 6–8 mg, the temperature range was 25 °C to 800 °C, the heating rate was 10 °C/min, and the protective gas flow rate was 20 mL/min.

### 3.5. Characterization of Photothermal Conversion Properties

The untreated silk fabric, the silk fabric treated with the silane coupling agent KH570, and the silk fabric cotreated with different concentrations of C@SiO_2_ nanoparticle solutions were exposed to 808 nm near-infrared light and sunlight, and the changes in the silk fabric were observed via thermal imaging with increasing irradiation time and temperature.

### 3.6. Washing Durability Test

The photothermal properties of the samples were tested according to FZ/T73023-2006 [39] “Antimicrobial Knitwear” by performing 10, 30, and 50 standard washes to assess the washing resistance of the photothermal effect of the finished silk fabric.

### 3.7. UV Resistance Test

The UV transmittance of the fabric was tested using a UV spectrophotometer (Shimadzu UV-2600, Shimadzu, Kyoto, Japan) with a slit width of 2 nm in the range of 190–800 nm. The transmittance was measured by first using barium sulfate as a base for baseline calibration and then using a sample holder to hold the sample under test and scanning at wavelengths 190–800 nm. According to GB/T18830-2009 [40] “Evaluation of textile UV protection performance”, the UV protection factor and UV transmittance T (UVA) and T (UVB) of the samples were measured at 290–400 nm using a UV protection factor (UPF) tester.

## 4. Conclusions

In conclusion, we successfully prepared C@SiO_2_ nanoparticles by coating silica on the surface of carbon nanoparticles, and we modified the surface of silk with C@SiO_2_ nanoparticles through grafting via chemical bonding with the silane coupling agent KH570. By comparing the transmission electron microscopy (TEM) and infrared spectroscopy (FTIR) results of the carbon and C@SiO_2_ nanoparticles, we found that SiO_2_ was successfully modified on the surface of the carbon nanoparticles to form nanoparticles with a core–shell structure. By comparing the absorbance of carbon and C@SiO_2_ nanoparticles in the UV–Vis–NIR spectra and the UV absorbance at different time periods in aqueous solution, we found that C@SiO_2_ nanoparticles were more dispersed and stable than carbon nanoparticles in aqueous solution. The infrared spectroscopy (FTIR) and energy dispersive spectrometer (EDS) results showed that C@SiO_2_ nanoparticles were successfully used to modify the surface of silk. The results of the photothermal conversion experiments demonstrate that C@SiO_2_ silk fabrics possess suitable photothermal conversion properties. To determine the strength of the bond of the C@SiO_2_ nanoparticles to the silk fabric, we tested the C@SiO_2_ silk fabric after repeated rinsing. We found that the photothermal effect of C@SiO_2_ silk fabric remained virtually unaltered after various washes, proving that the nanoparticles were firmly bonded to the silk fabric. The UPF value of the C@SiO_2_ silk fabric was calculated by measuring the UV transmission rate of C@SiO_2_ silk fabric, which showed good UV resistance. In this study, low-cost, thermally efficient, and environmentally friendly C@SiO_2_ silk fabrics were produced. This material shows potential for future application in the textile industry.

## Data Availability

Data are contained within the article.

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
