# Peer review of "The Photothermal Conversion and UV Resistance of Silk Fabrics Being Achieved through Surface Modification with C@SiO2 Nanoparticles"

_molecules, 2023, doi:10.3390/molecules28247970_

Round 1

Reviewer 1 Report

Comments and Suggestions for Authors

The authors report the protocol of modification of the silk fibers with C@SiO2 nanoparticles and their application as photothermal material. The high stability of the material is demonstrated.

The main drawback of the manuscript is the absence of the comparison the properties with other known materials to demonstrate the advantage of reporeted one.

Author Response

Comparison with other fabric heating materials has been added to Table 2

Techniques

Materials

Heating

performance

Ref.

Photothermal

MoS2-HNSPs

fabrics

25 ℃ to 52 ℃ after

irradiation 60 s

[16]

CNT/cellulose

30 ℃ to 40 ℃ after

irradiation 60 s

[19]

CNF/Ti3C2Tx

20 ℃ to 40 ℃ after

irradiation 200 s

[38]

NM/PDMS

fabrics

38.4 ℃ to 45.3 ℃ after

irradiation 300 s

[6]

C@SiO2

fabrics

23 ℃ to 60 ℃ after

irradiation 30s

Our

work

Reviewer 2 Report

Comments and Suggestions for Authors

The manuscript represents a development of a method for modifying the silk surface with C@SiO2 nanoparticles. The paper topic is completely covering the scope of the Molecules journal, paper conclusions are certainly interesting for the readership of the journal. The paper is structured, the material is presented in a consistent way, all conclusions are justified and supported by the results, results are presented in detail and accurately, and the English language is appropriate and understandable. The authors have successfully developed a technique for modifying silk fibre to improve its photothermal and UV-protective properties. The work provides an advance towards the textile industry, which makes it publishable in the Materials journal upon some minor corrections.

1.     No evidence is given that transparent layer in the TEM images is SiO2. EDX mapping or HRTEM should be done. TEM images are obtained in dark field mode? Please specify.

2.     How does the particle charge change after modification of carbon nanoparticles with silicon dioxide? The authors write about the increase in the number of hydroxyl groups on the surface of particles. This should lead to an increase in charge and stabilisation of particles, but the opposite is observed.

3.     In the discussion of the IR spectra, the 1630 cm-1 band refers to the bending of water rather than Si-O.

4.     On page 7, the positions of some IR bands are presented with too great precision, down to 0.1 cm-1. This is incorrect because the resolution of the acquisition was 4 cm-1.

Author Response

The EDS experiment in question 1 has been supplemented with the measurement of SI and O elements, proving that carbon is wrapped around a layer of silica.
The ZETA experiment in question 2 has been added, which demonstrated that the negative charge on the surface of carbon nanomaterials encapsulated in silica increases, making the nanoparticle dispersion more stable.
The infrared spectroscopy error in question three has been removed.
Errors in the infrared spectra of question four have been removed

Reviewer 3 Report

Comments and Suggestions for Authors

In the article entitled “The Photothermal Conversion and UV Resistance of Silk Fabrics Achieved Through Surface Modification with C@SiO2 Nanoparticles,” Deng et al. report the synthesis of carbon nanoparticles coated with SiO2. These nanoparticles have been attached to the surface of silk fabric using KH570 as a coupling agent. This nanocomposite material can be used as a smart textile due to its competitive photothermal properties and UV resistance of silk fabrics. The article is coherent, and the conclusions are well supported by experimental data. However, some points should be addressed before its publication:

  • Some errors in the English language should be corrected.
  • The crystalline phase of the carbon nanoparticles is not defined. Are they amorphous, graphite, or nanodiamonds?Provide experimental evidences.
  • There are several studies in the literature that report the use of carbon nanoparticles and carbon nanotubes in smart textiles. Please compare your results with the previously reported ones.
  • It is suggested to conduct a study on the nanotoxicity of this material and its potential impact on the environment after its useful life has ended.

Comments on the Quality of English Language

There are some grammatical and punctuation errors.

Author Response

The language has been touched up.
The crystalline phase of the carbon nano has been determined by XRD to be graphitic carbon.
Properties have been compared with other similar materials.

Round 2

Reviewer 1 Report

Comments and Suggestions for Authors

The authors have improved the manuscript in accordance with the comments and it can now be published.